# The use of 3D digital anatomy model improves the communication with patients presenting with prostate disease: The first experience in Senegal

**Babacar Diao**[1,2]*, **Ndèye Aissatou Bagayogo**[1], **Nayra Pumar Carreras**[3], **Michael Halle**[4], **Juan Ruiz-Alzola**[3], **Tamas Ungi**[5], **Gabor Fichtinger**[5], **Ron Kikinis**[4]

1 Department of Urology, Ouakam Military Hospital, Dakar, Senegal, 2 Faculty of Medicine Sheikh Anta Diop University, Dakar, Senegal, 3 Research Institute in Biomedical and Health Science, University of Las Palmas de Gran Canaria, Las Palmas, Spain, 4 Department of Radiology, Surgical Planning Laboratory, Brigham and Women's Hospital, Harvard Medical School, Boston, MA, United States of America, 5 Laboratory for Percutaneous Surgery, School of Computing, Queen's University, Kingston, Canada

* babacar.diao@ucad.edu.sn

**Data Availability Statement:** All relevant data are within the paper and its Supporting Information files.

## Abstract

### Objectives

We hypothesized that the use of an interactive 3D digital anatomy model can improve the quality of communication with patients about prostate disease.

### Methods

A 3D digital anatomy model of the prostate was created from an MRI scan, according to McNeal's zonal anatomy classification. During urological consultation, the physician presented the digital model on a computer and used it to explain the disease and available management options. The experience of patients and physicians was recorded in questionnaires.

### Results

The main findings were as follows: 308 patients and 47 physicians participated in the study. In the patient group, 96.8% reported an improved level of understanding of prostate disease and 90.6% reported an improved ability to ask questions during consultation. Among the physicians, 91.5% reported improved communication skills and 100% reported an improved ability to obtain patient consent for subsequent treatment. At the same time, 76.6% of physicians noted that using the computer model lengthened the consultation.

### Conclusion

This exploratory study found that the use of a 3D digital anatomy model in urology consultations was received overwhelmingly favorably by both patients and physicians, and it was perceived to improve the quality of communication between patient and physician. A

**Funding:** The authors received no specific funding for this work.

**Competing interests:** The authors have declared that no competing interests exist.

randomized study is needed to confirm the preliminary findings and further quantify the improvements in the quality of patient-physician communication.

## Introduction

Prostate diseases, represented by benign prostate hyperplasia (BPH) and prostate cancer, are the leading causes of urinary disorders of the lower tract with varying incidences depending on age, geographic regions, and ethnicity [1–4]. BPH is the most common of such diseases, causing high morbidity and significantly impaired quality of life. Its prevalence increases with age and, as confirmed by studies of postmortem pathology samples, reaches 70% beyond 60 years of age [5, 6]. Prostate cancer (PCa) is the most frequent cancer and the leading cancer-related cause of death in African men [7]. Early-stage PCa is often asymptomatic or shares symptoms of benign prostatic hyperplasia. While communication campaigns in Western countries have led to better knowledge and increased awareness of prostate diseases [8], but in sub-Saharan Africa, such communication traditions and techniques are lagging which makes it extremely difficult to manage patients in diagnostics, therapy, and follow-up. Patients tend to have limited knowledge of prostate diseases and are often not seen until presenting complicated or advanced disease, leading to poor therapeutic outcomes [9–11]. Due to lack of screening and public awareness of prostate diseases in Africa, PCa is discovered at advanced stages [12]. Sub-Saharan countries follow the same trend, although reliable national statistics on prostate diseases are scarcely available due to a lack of adequate recordkeeping [13]. In Senegal, over 60% of patients present with advanced cancer and only less than 40% with treatable localized disease [14–16], a stark contrast to the USA where about 89% of PCa are diagnosed early [17]. Alas, the five-year survival rate after diagnosis hovers around 30% in Senegal [16], versus 98% in the USA [17].

This paper concerns physician-patient communication consultation in urology setting, specifically regarding prostate disease. It has been well understood that effective patient-physician communication can improve a patient's health quantifiably, and in turn, it reduces healthcare costs [18], Patients who understand their doctors are more likely to acknowledge health problems, understand their treatment options, modify their lifestyle and behavior accordingly, and follow prescribed treatment regimen; ultimately leading to more favorable clinical outcomes [19].

We hypothesized that the use of 3D digital anatomy models would be received favorably by both physicians and patients, and in turn, it would improve the quality of communication during consultation between physicians and patients presenting with prostate disease. The purpose of this exploratory study was to demonstrate the technical feasibility of using 3D digital anatomy models in Senegal and gauge the perception of this technology by both physicians and patients and its potential impact of this technology on the quality of communication between physicians and patients.

## Methods

### 3D digital anatomy model

The 3D digital anatomy model of the prostate was based on McNeal [20, 21] zoning that divides the prostate into five areas where tumors usually develop: peripheral zone where 70% of prostate cancers grow; transition zone where BPH develops and 30% of cancers grow; central zone with the ejaculating channels; peri-urethral zone where BPH initiates; and anterior

fibromuscular zone. The urethra and rectum were included to help explain the nature of impaired urinary flow and digital rectal examination (DRE), respectively.

For this exploratory study, we selected a single representative case, deemed similar many of the cases seen in initial urology consultation in Senegal. Multiparametric MRI of a 62 y.o. man, presenting with urinary disorder of the lower tract, PSA over 13 ng/ml, negative biopsy, was acquired, with 1.5T Siemens scanner, 3mm slice thickness. The MRI showed PI-RADS 1 lesions with an enlarged transition zone. The MRI scan was imported in DICOM format to the 3D Slicer free open-source medical image visualization software [22, 23], running on a laptop. The McNeal zones, urethra, and rectum were manually contoured (Fig 1).

The resulting 3D digital anatomy model (Fig 2) was saved for subsequent viewing in 3D Slicer on a laptop, tablet, or desktop computer, with convenient functions to zoom in/out, rotate and selectively show/hide structures, similarly to most other existing digital anatomy viewers [24]. The participating physicians received an introductory training on the use of the 3D Slicer interactive software to visualize the 3D digital anatomy model on their own office computer.

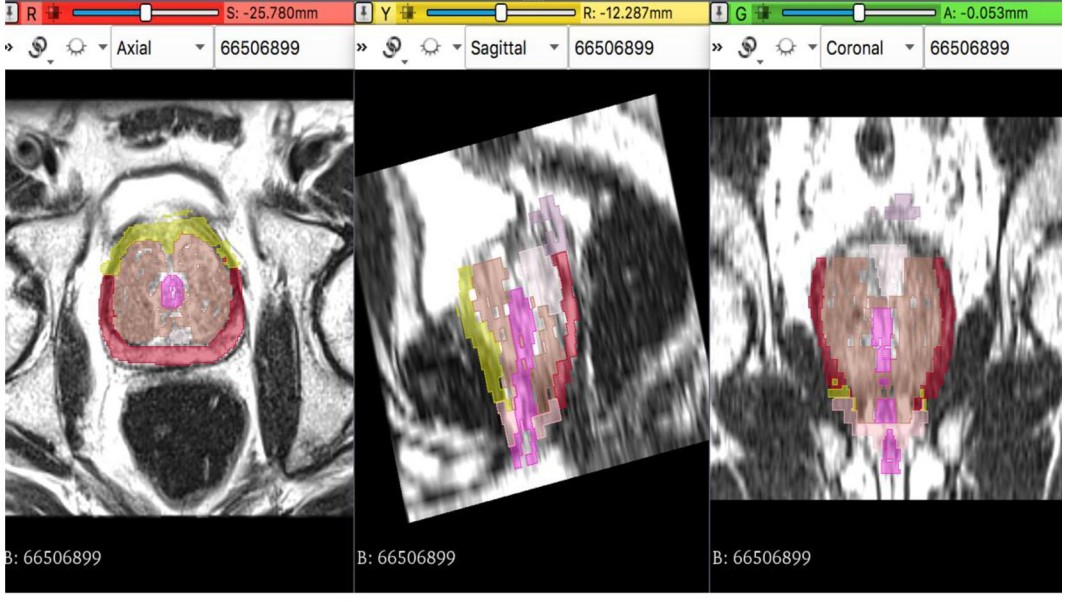

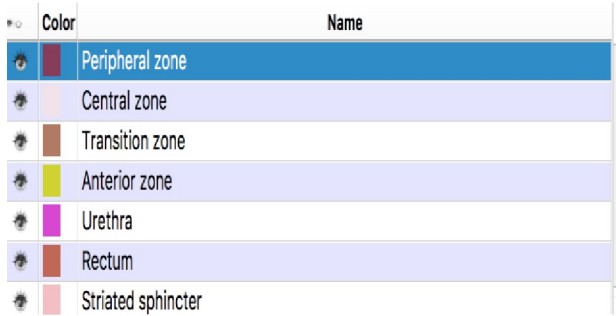

**Fig 1. MRI images with segmentation of labels, in the mid-section of the prostate.**

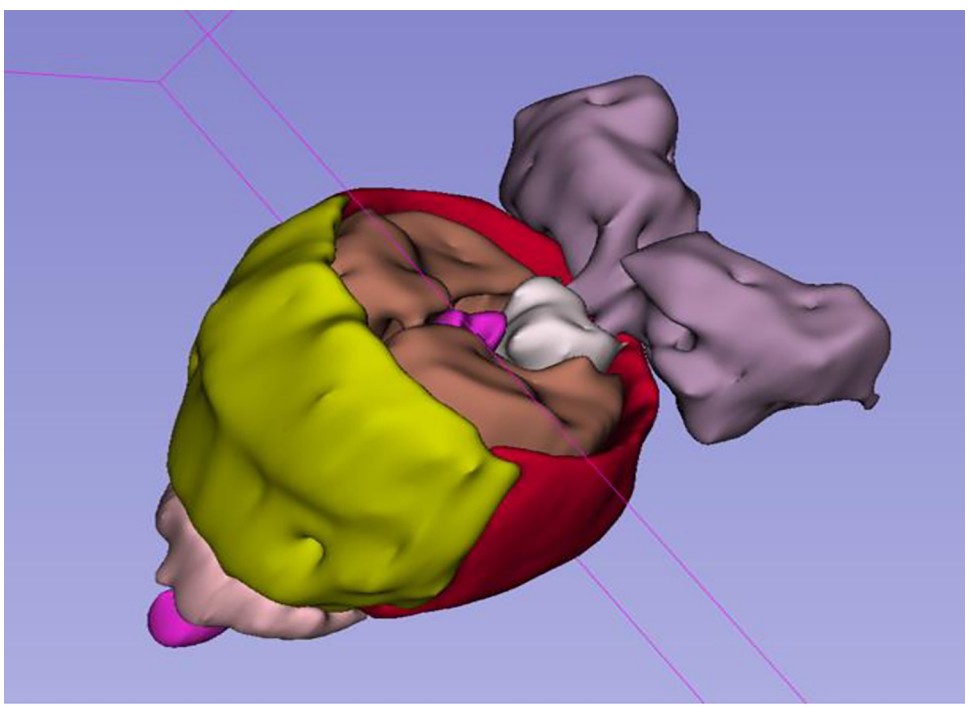

| | Color | Name |
|---|---|---|
| | | Peripheral zone |
| | | Central zone |
| | | Transition zone |
| | | Anterior zone |
| | | Urethra |
| | | Rectum |
| | | Striated sphincter |

**Fig 2. Screen capture of the 3D digital anatomy model, showing McNeal zones.**

## Study design

We conducted a single-arm prospective qualitative study of preliminary and exploratory nature, to gauge technical feasibility and the perception of this technology by both physicians and patients.

**Table 1. Questionnaire for patients (with 95% confidence intervals (CI) from/to).**

| Questionnaire for patients | Numbers | Percent | CI from | CI to |
|---|---|---|---|---|
| Q1. How was your level of understanding impacted by the computer model? | | | | |
| • Improved | 298 | 96.8% | 95% | 99% |
| • No change | 10 | 3.2% | 1% | 5% |
| • Decreased | 0 | 0% | | |
| Q2. Was your decision about treatment impacted? | | | | |
| • Yes | 222 | 72% | 67% | 77% |
| • No | 86 | 28% | 23% | 33% |
| Q3. If yes, which method impacted your decision? | | | | |
| • Computer model | 194 | 87.4% | 84% | 91% |
| • Manual drawing | 28 | 12.6% | 9% | 16% |
| Q4. Did computer model help you ask more questions? | | | | |
| • Yes | 279 | 90.6% | 87% | 94% |
| • No | 29 | 9.4% | 6% | 13% |

The study involved patients referred for urological consultation, presenting with symptoms or concerning findings in the lower urinary tract, attributed to suspected prostate disease. The physician group consisted of attending staff and clinical residents specializing in urology or general medicine, experienced in consulting patients presenting with prostate disease. During the urology consultation, each patient first received an explanation of his suspected prostate disease using conventional means, including manual drawings and verbal explanation, as usually done by the given physician. We did not provide a "standardized" drawings or didactic images for the physicians who conducted the consultation according to their own usual practice. Next, informed consent was sought from the patient for participation in the study. If the patient opted in to participate, a second explanation of the suspected disease followed, this time using the 3D digital anatomy model rendered on the computer. At the end of the consultation, the patient was asked to fill out a straightforward questionnaire, shown in Table 1, about his overall experience with the quality of the consultation. Upon completing the study, each participating physician was asked to fill out a similarly straightforward questionnaire, shown in Table 2, about their own experience with consultation and the effects of using 3D digital anatomy model.

**Table 2. Questionnaire for physicians (with 95% confidence intervals (CI) from/to).**

| Questions for physicians | Numbers | Percent | CI from | CI to |
|---|---|---|---|---|
| Q1. Did your communication skill improve by using the computer model? | | | | |
| • Yes | 43 | 91.5% | 88% | 95% |
| • No | 4 | 8.5% | 5% | 12% |
| Q2. Did obtaining consent to treatment improve by using the computer model? | | | | |
| • Yes | 47 | 100% | 100% | 100% |
| • No | 0 | 0 | 0% | 0% |
| Q3. Was the length of communication reasonable with using the computer model? | | | | |
| • Yes | 11 | 23.4% | 19% | 28% |
| • No, it takes too long | 36 | 76.6% | 72% | 81% |
| Q4. Are you likely to encourage your colleagues to consider using the computer model? | | | | |
| • Yes | 47 | 100% | 100% | 100% |
| • No | 0 | 0 | 0% | 0% |

The questions were purposefully designed to be short and straightforward, for two main reasons. First, the participating patients were members of the general public, with varying and typically limited educational background; we tried to avoid "participant fatigue" and not to overwhelm the patient. Secondly, we tried not to increase excessively the overall length of consultation.

The questionnaires, as all documents in relation to this trial, were originally written in French and then translated to English for the purpose of this article.

## Statistical analysis

Confidence intervals were computed as follows. With confidence level of 0.95, alpha = 0.025, so Z = 1.96. The formula for confidence interval is: p-1.96*sqr(p*q/n) and p+1.96*sqr(p*q/n), where p is the sample proportion, q = 1-p, and n is the sample size.

Chi-square test of independence was performed to analyze the relation between referral groups and different responses in the questionnaire. Bonferroni correction was applied to compensate for repeated tests.

## Research ethics approval

This study was approved by the institutional human research ethics board of the Ouakam Military Hospital, Dakar, Senegal, as the principal performance site with additional satellite clinics. Participating patients gave signed consent, stating their understanding that the study was voluntary and conducted anonymously for research purposes. Patients were assured that no aspect of the study was included in their medical record. No specific incentives were offered to study participants, neither patients nor providers. During the informed consent process, however, patients were informed about potential benefits, such as receiving more detailed information about their disease and condition and additional "face time" with their consulting physician.

## Study period

This study took place between March 1, 2019 and June 30, 2019.

## Referral groups

A total of 47 physicians took part. Enrollment was offered to 387 patients, of whom 308 gave informed consent and participated in the study.

Each participating patient belonged to one of the following major referral groups:

- *First-time consultation*: 106 patients were seen for a first-time consultation, for whom it was necessary to explain the benefits and nature of DRE.

- *Biopsy candidates*: 74 patients were seen because it was recommended that they have a prostate biopsy based on previous DRE and/or high PSA.

- *Watchful waiting candidates*: 37 patients were seen because they had had a previous negative core needle biopsy of suspected cancer, for whom it was necessary to explain why they were not guaranteed to be free of cancer and why they needed regular follow-ups.

- *Prostatectomy candidates*: 29 patients were seen for positively diagnosed prostate cancer and who had been recommended to have radical prostatectomy.

- *TURP candidates*: 62 patients were seen for BPH with associated urinary symptoms, who were likely to require eventual surgical management.

Patient demographics (age, etc.) was not collected in this exploratory study.

## Results

### Answers to the questionnaires

In all, 308 patients and 47 physicians participated and filled out their questionnaires. The mean age of patients was 69.6 ± 6.7 years (46–98 years). Patient and physician experiences with the 3D digital anatomy model were reported in Tables 1 and 2, respectively.

Patient answers (see Table 1):

**Q1**: 96.8% of patients found that their level of understanding of the disease and subsequent treatment options had improved, while only 3.2% reported no change in understanding. No patient reported a decrease.

**Q2**: 72% of patients found that their decision regarding further management of the disease was impacted by the consultation, and (**Q3**) 87.4% of these patients found that the use of the 3D digital anatomy model impacted their decision, versus 12.6% who found the manual drawing to be more impactful.

**Q4**: 90.6% of patients found that the 3D digital anatomy model improved their ability to ask questions about treatment and follow-up.

Physician answers (see Table 2):

**Q1**: 91.5% of physicians found that the use of the 3D digital anatomy model improved their communication skills, versus 8.5% who reported it to be unaffected.

**Q2**: 100% of the physicians found that their ability to obtain patient consent for subsequent treatment and follow-up improved by using the 3D digital anatomy model.

**Q3**: 76.6% of the physicians felt that the use of the 3D digital anatomy model added too much time to the consultation.

**Q4**: 100% of the physicians were likely to encourage their colleagues to consider using 3D digital anatomy models.

### Statistical considerations

All answers given by 308 patients and 47 physicians to all questions, excluding one, were decisively positive; the single negatively answered question had over 76% majority. In this light, succinct descriptive statistics were deemed satisfactory for this exploratory study. On average, each physician saw 6–7 patients, a sufficiently large enough pool to capture the essence of their experiences in a statistically suggestive manner.

The 95% confidence intervals for the patient and physician questionnaires are provided in right-most columns of Tables 1 and 2, respectively.

Reported in Table 3, we broke down the patient answers by referral groups. At our sample size and significance level of 5%, we did not find significant differences between referral groups in answering Q1. Patients who visit for their first consultation found that their treatment decision was less often impacted (Q2), the computer model was less impactful in their decisions (Q3), and the computer model less often helped them ask more questions (Q4) compared to patients who come for a follow-up visit (p<0.001 for Q2, Q3, and p = 0.04 for Q4).

## Discussion

The most important finding is that, in our experience, the use of a 3D digital anatomy model was perceived to improve the quality of communication for both patients and physicians in all aspects addressed by the questionnaires.

**Table 3. Questionnaire for patients, broken down by referral groups.**

| | First consultation (106) | Biopsy candidates (74) | Watchful-waiting candidates (37) | Prostatectomy candidates (29) | TURP candidates (62) | Chi$^2$ p value |
|---|---|---|---|---|---|---|
| Q1. How was your level of understanding impacted by the computer model? | | | | | | p = 0.69 |
| • Improved (298) | 101(95.3%) | 72(97.3%) | 37(100%) | 28(96%) | 60(96.8%) | |
| • No change (10) | 5(4.7%) | 2(2.7%) | 0 | 1(3.4%) | 2(3.2%) | |
| • Decreased | 0 | 0 | 0 | 0 | 0 | |
| Q2. Was your decision about treatment impacted? | | | | | | p<0.01 |
| • Yes (222) | 59(55.7%) | 66(89.2%) | 26(70.3%) | 23(79.3%) | 48(77.4%) | |
| • No (86) | 47(44.3%) | 8(10.8%) | 11(29.7%) | 6(20.7%) | 14(22.6%) | |
| Q3. If yes, which method impacted your decision? | | | | | | p<0.01 |
| • Computer model (194) | 39(66.1%) | 63(95.5%) | 26(100%) | 23(100%) | 43(89.6%) | |
| • Manual drawing(28) | 20(33.9%) | 3(4.5%) | 0 | 0 | 5(10.4%) | |
| Q4. Did computer model help you ask more questions? | | | | | | p = 0.04 |
| • Yes (279) | 88(83%) | 74(100%) | 33(89.2%) | 29(100%) | 55(88.7%) | |
| • No(29) | 18(17%) | 0 | 4(10.8%) | 0 | 7(11.3%) | |

## Impact on patients

Improvement in the quality of communication from the perspective of patients is reflected by the questionnaire in Table 1: better understanding explanations (Q1), positive impact on treatment decisions (Q2, Q3), and increased number of questions asked (Q4).

**Understanding explanations.** In Senegal, where patients have a low level of literacy, understanding the nature of prostate diseases seems to have been facilitated by the 3D digital anatomy model. In the long run, this should help patients to participate and stay engaged in their disease management. Over the past decade, several studies across Africa have revealed a low level of knowledge and awareness of prostate cancer [25–30]. A tool that facilitates the communication and understanding of prostate diseases could be beneficial in improving the quality of care and reducing healthcare costs. According to Bennett et al. [31], low literacy level can be a significant barrier to early diagnosis of prostate cancer among low-income patients in America, and it is safe to assume that this is true throughout sub-Saharan Africa. It follows that appropriate communication strategies would improve awareness of prostate disease and promote earlier diagnosis. This, in turn, would reduce costs associated with the complex treatment of advanced diseases.

**Impact on treatment decisions.** The discussion with the patient for consenting to treatment is especially important when they present with high-risk disease, such as prostate cancer. The need to consider and accommodate the patient's preferences for disease management has long been the cornerstone of communication between physician and patient [32] and the use of a 3D digital anatomy model seemed helpful in this regard.

Table 3 shows that, on the whole patients favored of the digital model regardless of the reference group. However, the differences were less significant in decision-making for first-time consultants and patients monitored by watchful waiting. *At the same time, the differences were much more significant in decision-making for "high-risk" patient groups, especially the biopsy candidate group, who needed immediate management of the disease.*

**Ability to ask questions.** Encouraging patients to ask well-informed questions should help them understand not only the benefits, but also the risks of treatment. The quality of

physician-patient communication has direct impact on managing expectations and post-treatment grievances [33]. In our study, the use of a 3D digital anatomy model allowed the patient to better understand the risks and potential consequences of procedures, whether biopsy, radical prostatectomy or TURP. With just a few simple mouse clicks on the computer, physicians were able to explain complicated concepts, like the mechanism of DRE as an important diagnostic tool, or what radical prostatectomy involves in relation to functional outcomes, or the need for regular follow-ups after a negative biopsy. Lack of understanding of basic health information has had a well-documented negative impact on health care [34] and thus, the importance of making advances in patient education cannot be underestimated. Our preliminary study suggests that a modest or low level of literacy and knowledge of prostate diseases is not an insurmountable obstacle, and that the gap can be bridged by improved communication, in which the use of a 3D digital anatomy model appeared to be decidedly helpful. In this regard, our findings are consistent with Rajbabu et al. [35], who found that low levels of income and literacy are associated with poor awareness of prostate cancer, but that this situation may be improved by effectively conveying information about the nature of the disease and its management.

The mean age of patients was 69.6 ± 6,7 years in this exploratory study. In Senegal, men younger than 55 years of age are seldom seen in urology consultation. Practically speaking, nearly all patients who participated in our study were born before the "digital age", and they were likely to have been affected similarly by the "novelty factor" of digital anatomy technology, regardless of their exact age.

## Impact on physicians

The benefits of using a 3D digital anatomy model, from the perspective of physicians, is reflected by their answers to the questionnaire shown in Table 2: improved communication skills (Q1), improved ability to obtain consent to treatment (Q2), and high likelihood of encouraging colleagues to adopt this new technology (Q4), while criticism was expressed for lengthening the consultation (Q3).

**Communication skills.**   Communication techniques are not taught in medical school in Senegal. The use of an interactive 3D digital anatomy model represents not only innovation but is also a tool to guide and help the physician in conveying information to the patient, thus facilitating communication. It can also be a useful tool for the physician to emphasize certain aspects of the disease and to finetune communication of the patient's specific needs in consideration of his symptoms and prostatic pathology.

**Obtaining consent to treatment.**   Our findings were consistent with existing literature. Our 3D digital anatomy model can be considered conceptually similar to the cartoon illustrations proposed by Delp et al. [36], which were found to improve adherence to treatment. Haskard Zolnierek et al. [37] found that the risk of non-adherence to treatment by patients increases by 19% when the physician does not communicate well with the patient, and the chances of adherence are 1.6 times higher when the physician has undergone communication training.

**Lengthening the consultation.**   The increased length of the consultation, which was reported by the majority of physicians, is a significant concern. Simple didactic anatomic pictures play a similar role and they take less time. During the first part of the consultation session the physicians used some didactic anatomical pictures of their own design and drawing. Nonetheless, both physician and patients uniformly reported favorable experience from additionally using 3D digital anatomy models. Hence in this paper we argue that the use of 3D digital models may help improve upon the quality of patient-physician communication.

This could be explained by that fact that using interactive 3D anatomy model is a novelty for both parties. Over 90% of patients found that the 3D digital anatomy model improved their ability to ask questions, and more questions always lead to longer consultations. For physicians, the 3D digital anatomy model is a new technology that involves a learning curve; it is quite probable that the length of each consultation would be reduced as physicians become more proficient in using this technology on the computer. Moreover, the use of the 3D Slicer software adds complexity and can be confusing for novice users and may require extra time to use during consultations. However, one can argue that, in contrast to the current practice, the increase in the length of consultations may be well-compensated by the gain in the quality of communication, and in turn, in the quality of care.

**Encouraging colleagues to adopt the technology.**  The willingness, unanimously expressed by physicians in this study, to popularize the use of 3D digital anatomy models reinforces the hope that this technology will be well-appreciated by its physician users.

## Limitations

**Study design.**  In this single-arm exploratory and preliminary study, with each patient we conducted two urological consultations, back-to-back: first a conventional consultation session that was immediately followed by a session using 3D digital anatomical models, and then we gauged the perceived improvement on the quality of the physician-patient communication through simple questionnaires.

The ultimate objective of improving physician-patient communication is to improve patient willingness to pursue further care and sustained compliance with the prescribed follow-up regimen, a crucially important aspect for all patients with prostate disease. Although this could not be measured in this short preliminary study, existing literature supports our expectation that improved patient-physician communication should lead to improved patient compliance in general [37]. In follow-up work, we plan to run a randomized study with two arms of patients, one receiving conventional consultation and one receiving consultation with using 3D digital anatomical models. This future study would allow us to compare clinical outcome variables, such as willingness to pursue further care, etc.

**Choice of questionnaires.**  We used simple binary questions to gage various aspects in patient and physician experience with the 3D digital anatomy model, with respect to the quality of communication during consultation. We considered the use of scaled feedback, even as simple as Likert scales, and decided not to follow that approach. As our patients have a very modest level of literacy, they could have been easily overwhelmed by more complicated questions involving concepts like "scale" or "percentage". The physicians, however, could have been asked more nuanced questions, and this will be considered in future studies.

In this exploratory study, we intended to size up the perception of participant, rather than more objectively assessing knowledge and experience gained by the participants through using of 3D digital anatomy models and whether the actual communication skills of the physician is improved, for which we will need to capture and analyze what is actually said and communicated to the patients; all this, however, we were compelled to reserve for future work. In further follow-up work, in the randomized dual-arm study mentioned earlier above, we would construct the questionnaires differently and would also break down the results by referral groups; it is suspected that patients with benign and malignant conditions may react to the outcome of urological consultation differently and thus may differ in their willingness to pursue further care.

**Choice of the 3D anatomical model.**  The 3D digital model was not personalized based on the individual patient case, for this is not possible in Senegal, for multiple reasons. For

many patients, the consultation session studied in this paper was the first meeting between the physician and the patient upon referral, when practically no Senegalese patient has prior imaging scan of any sort. Moreover, due to general lack of imaging capacity, especially that of MRI. In future work, we consider creating several anatomical models, and based on a quick digital rectal examination, the physician could select the one that feels the most similar one to the prostate of the given patient.

**Choice of digital anatomy viewing software.** During the urology consultation, the 3D digital anatomy model was viewed with the free, open-source 3D Slicer software [22, 23]. 3D Slicer is an extremely powerful and complex research platform for medical image analysis and visualization, offering over 1,000 major functions in its core, and fortified by over 100 extension packages. The use of 3D Slicer for the sole purpose of visual rendering may have been overly complex for new users, and it was probably the most important reason why over 76% of the physicians felt that using the 3D digital anatomy model lengthened the consultation exceedingly. In future work, we consider using the Open Anatomy Browser [22], another free open-source digital anatomy viewing platform that is much simpler to learn and operate, which, in turn, should help reduce the length of the consultation session without loss of effectiveness. Moreover, the Open Anatomy Browser visualizes 3D digital anatomy model created in 3D Slicer, allowing is to reuse previously created models. The Open Anatomy Browser works through a public web interface, accessible from mobile device, such as smart phone, that practically all Senegalese physicians possess. Finally, the simple user interface of the Open Anatomy Browser may allow patients to review on their own mobile device the main points of the consultation with their family and their referring physician.

**Robustness of findings over time.** One significant limitation of this study was the 4-month duration, a time frame insufficient to ascertain the consistency and robustness of our findings over time, particularly of the enthusiasm of doctors to use the 3D digital anatomy model and their desire to popularize it among their colleagues. It is possible that part of their enthusiasm was generated by the appealing technological novelty and the fact this study was the first of its kind in Senegal and, according to our knowledge, in sub-Saharan Africa.

**Relationship to public awareness of prostate disease.** The work presented here is not a "community intervention" on awareness, but rather can be considered as a "clinical intervention" following initial referral for consultation. Increasing the awareness of prostate disease and improving the management of the disease (including patient-physician communication, the subject of this paper) must go hand in hand. In Senegal, we promote the awareness of prostate disease to the public through the annual Prostate Disease Awareness Day (PDAD), a sweeping outreach effort in the Senegalese media and social platforms. Participating media outlets include the Senegalese National Television, 4 private television networks, 6 radio stations, and 2 major social platforms. PDAD provides not only general information to the public, but also free, walk-in urology consultation for ad-hoc patients. In Dakar alone, over 60 participating physicians (urologists and general practitioners) provide free clinical consultation during PDAD.

## Conclusion

In this exploratory study, we found that the use of a 3D digital anatomy model in urology consultations was received favorably by both physicians and patients. It was perceived to improve the quality of communication between patient and physician, and both parties reported an overwhelmingly positive experience. Further studies are required to quantify these preliminary qualitative findings. Although this study concentrated on prostate diseases, the same technology should translate to other important areas, such as women's and maternal health, which are among the most pressing health problems in sub-Saharan African countries.

## Supporting information

**S1 Data.**
(XLS)

**S2 Data.**
(XLS)

## Author Contributions

**Conceptualization:** Babacar Diao, Juan Ruiz-Alzola, Gabor Fichtinger, Ron Kikinis.

**Formal analysis:** Babacar Diao, Tamas Ungi, Gabor Fichtinger, Ron Kikinis.

**Investigation:** Babacar Diao.

**Methodology:** Babacar Diao, Nayra Pumar Carreras, Michael Halle, Juan Ruiz-Alzola, Gabor Fichtinger, Ron Kikinis.

**Project administration:** Ndèye Aissatou Bagayogo.

**Supervision:** Juan Ruiz-Alzola, Gabor Fichtinger, Ron Kikinis.

**Visualization:** Nayra Pumar Carreras, Michael Halle.

**Writing – original draft:** Babacar Diao, Ndèye Aissatou Bagayogo.

**Writing – review & editing:** Babacar Diao, Gabor Fichtinger.

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
