## [Decision Letter · Decision Letter 0]

20 Jun 2022

PONE-D-21-12063The use of 3D digital anatomy model improves the quality of communication with patients presenting with prostate disease, first experience in SenegalPLOS ONE

Dear Dr. Diao,

Thank you for submitting your manuscript to PLOS ONE. After careful consideration, we feel that it has merit but does not fully meet PLOS ONE’s publication criteria as it currently stands. Therefore, we invite you to submit a revised version of the manuscript that addresses the points raised during the review process.

Your manuscript has been assessed by three expert reviewers, whose comments are appended below. The reviewers have highlighted concerns about several aspects of the methodology and discussion. Please ensure you respond to each point carefully in your response to reviewers document, and modify your manuscript accordingly.

We look forward to receiving your revised manuscript.

Kind regards,

Joseph Donlan

Editorial Office

PLOS ONE

Journal Requirements:

2. Please provide additional details regarding participant consent. In the ethics statement in the Methods and online submission information, please ensure that you have specified whether consent was informed.

3. Thank you for stating the following in the Acknowledgments Section of your manuscript: "Financial support:  U.S. NIH/NIBIB 5P41EB015902-22, U.S. NIH/NIBIB 3P41EB015902-21A1S1, U.S. NIH/NIMH 1R01MH112748-03, BWH Precision Medicine Pilot Research Award, EU INTERREG MAC Program (MAC/1.1b/098), Natural Sciences and Engineering Research Council of Canada, U.S. NIH/NIBIB P41EB015898-15, U.S. NIH/NCI 1R01CA235589-01A1."

Please remove any funding-related text from the manuscript and let us know how you would like to update your Funding Statement. Currently, your Funding Statement reads as follows: "The authors received no specific funding for this work"

Reviewers' comments:

Reviewer's Responses to Questions

**Comments to the Author**

1. Is the manuscript technically sound, and do the data support the conclusions?

Reviewer #1: Yes

Reviewer #2: No

Reviewer #3: Partly

2. Has the statistical analysis been performed appropriately and rigorously? 

Reviewer #1: N/A

Reviewer #2: No

Reviewer #3: No

3. Have the authors made all data underlying the findings in their manuscript fully available?

Reviewer #1: Yes

Reviewer #2: Yes

Reviewer #3: No

4. Is the manuscript presented in an intelligible fashion and written in standard English?

Reviewer #1: Yes

Reviewer #2: Yes

Reviewer #3: Yes

5. Review Comments to the Author

Reviewer #1: The manuscript is simple but informative. Also the results are expected.

The number of patients and physicians accessed was reasonable.

While any image is better than no image, I would add simple anatomic figures to be more comprehensive to patients.

It is a huge challenge the fact that "76.6% of physicians complained that using the computer model lengthened the consultation."

This might reflect the complexity and inaccuracy of utilized model and interface (computer).

In our practice, simple and didactic anatomic pictures on the physician's office wall play similar role, certainly taking less time.

Reviewer #2: This manuscript presents the use of a 3D digital anatomy model of the prostate to help improve patient-provider communication in Senegal. While the concept is quite interesting, this study lacks depth and clarity. This manuscript could be strengthened with major revisions. Below, are outline comments by sections.

Introduction

• Should identify more current statistics on worldwide rates of prostate cancer deaths.

• It is unclear how “sociocultural realities that render prostate disease a taboo subject” is ameliorated or even considered in the context of this intervention. It is also unclear how this intervention has the potential to impact limited knowledge of prostate disease that are often not seen until presented complicated or as advanced disease. This is not a community intervention on awareness, but rather a clinical intervention after clinical presentation.

• This authors must reconsider what is important to present as relevant background information for this study.

• Given the preliminary and exploratory nature of this study, this should be stated clearly upfront.

Methods:

• Given the discussion of the 3D digital anatomy model section, it is unclear if a standard image was shown to all patients in the study, or if the image was personalized based on the individual patient case. It is unclear why there is a section on a specific patient case of a 62 year old man.

• Under study design, authors indicate “patient first received … standard manual drawings, followed by … 3D digital…” What is the standard manual drawing based on? Is it just what the physician was able to draw themselves or a pre-drawn out image on a paper? More detail is needed for clarity.

• The questions are quite limited in nature and should be addressed in a limitations section.

• What, if any, incentives were provided to study participants – whether patients or providers?

Results

• A comparison/usual care group could be much more informative and insightful, along with more detailed questions to assess quality of communication, understanding, communication skills, etc., even for low literacy groups, with low response burden.

• Authors should be more clear that the questions assess perception/perceived point of view. For example, asking about improved level of understanding as improved or declined, is perceived and not objective as asking actual questions that assess knowledge. This should also be acknowledged in a limitations section.

Discussion

• Authors should rephrase discussion section and conclusions drawn to be more accurate. For example, quality was not assessed and the conclusion that quality of communication improved is not accurate. There was no comparison group to show an improvement, nor a pre/post questionnaire, for example.

• Impact on physicians – There is also extreme over simplification of communication skills. Nothing that was presented in the intervention dealt with communication – e.g., what is actually said/communicated to patients. This 3D image is a visual “tool” that has the potential to enhance communication, but that was not measured in this study, but rather participants’ perspective on usability or utility, and even that is quite limited in the 4 questions asked of patients and providers.

• Statistical significance – It is unclear why there is a section on statistical significance when this study was exploratory in nature and only presented descriptive statistics. This section seems inappropriate.

• The additional subheadings following the discussion section is confusing, when some appear to be appropriate for the methods section.

• Digital software – what is the difference between the two platforms mentioned, besides one being free? Why bring this up as an option?

• The section on longevity of findings also seem inappropriate and inaccurate given the nature of this study.

Reviewer #3: The manuscript represent the result of a survey on the use of a 3D digital anatomy model in urology consultation to reflect its effect on improving the communication of physician and patients. An interesting model used in this study which based on the results of the survey demonstrated an improved communication between physicians and patients. However, I do not believe the survey results are presented and discussed properly. in more detail (e.g. consider results in referral groups and).

1. It is not clear if patients information such as age is collected in the survey. If it has been collected, the results should be reported and investigated in terms of any association with age groups (e.g. elderly patients might gain less benefit from the technology compared to younger patients). Otherwise, it should be mentioned that age of patients are not recorded in survey with a discussion on the reason.

2. There is no statistical method applied to investigate the result of the survey. Authors need to apply statistical methods to add the uncertainty associated with the reported percentage (e.g. in terms of confidence intervals). Also the chi-square or other relevant tests should be applied to investigate the association of each questionary with available patient groups (referral group, age groups, etc).

3. Authors need to report results of the questionaries in the survey by the referral groups and investigate any association between referral groups and each questionary (e.g. different treatment usually are recommended for patients with different group so it would be interesting to consider how patients’ decision has changed regarding their referral group).

4. Impact on patients in discussion, it is mentioned that “A tool that facilitates the communication and understanding of prostate diseases could be beneficial in improving the quality of care and reducing healthcare costs”. However, it is not well discussed how such a tool could reduce healthcare cost, and I suggest authors to add a discussion justifying this. (lines 168-170)

5. They mentioned that the use of such a 3D digital anatomy model is helpful for treatment decision of high-risk patients. It is not clear how the presented results are supporting such a statement! A clear discussion needs to be added including how the patients decisions are improved by use of such a 3D digital anatomy model. (lines 177-182)

6. Some of subsections in the discussion should be moved and presented in the Method section (i.e. Statistical significance, Choice of questionnaires and Choice of digital anatomy viewing software).

7. It is not clear how such a software would be available to be used in the clinical practices if required.

6. PLOS authors have the option to publish the peer review history of their article (what does this mean?). If published, this will include your full peer review and any attached files.

Reviewer #1: **Yes: **Leonardo Oliveira Reis

Reviewer #2: No

Reviewer #3: No

---

## [Author Response · Author response to Decision Letter 0]

13 Sep 2022

Response to the Reviews

We thank the editor and the reviewers for the helpful comments. Our point-by-point responses below detail the revisions.

Reviewer 1

The changes addressing these comments are marked by green color in the revised manuscript.

Comment: The manuscript is simple but informative. Also the results are expected.

Answer: Thank you for this positive comment. We were indeed glad to see that our results support the hypothesis. [No editing is required to address this comment].

Comment: The number of patients and physicians accessed was reasonable.

Answer: Thank you for this positive comment. [No editing is required to address this comment].

Comment: While any image is better than no image, I would add simple anatomic figures to be more comprehensive to patients…. In our practice, simple and didactic anatomic pictures on the physician's office wall play similar role, certainly taking less time.

Answer: We agree, while noting that in our study, during the first part of the consultation session the physicians used some didactic anatomical pictures of their own design and drawing. Nonetheless uniformly reported favorable experience from additionally using 3D digital anatomy models. Hence in this paper we argue that the use of 3D digital models may help improve upon the quality of patient-physician communication. We emphasize this in the revised text. [See in Methods / Study Design]

Comment: It is a huge challenge the fact that "76.6% of physicians complained that using the computer model lengthened the consultation. This might reflect the complexity and inaccuracy of utilized model and interface (computer). 

Answer: We agree that lengthening the consultation session is indeed a significant concern. The use of 3D digital models doubtless adds complexity which in turn adds time to the consultation. In the revised Discussion section, we propose ways to mitigate this effect, such as using a simpler digital anatomy viewing software. [See in Discussion / Impact on physicians]

Reviewer 2

The changes addressing these comments are marked by red color in the revised manuscript.

Comment: This manuscript presents the use of a 3D digital anatomy model of the prostate to help improve patient-provider communication in Senegal. While the concept is quite interesting, this study lacks depth and clarity. This manuscript could be strengthened with major revisions. 

Answer: We agree and carried out substantial revisions to improve on the depth and clarity of the paper. [See throughout the text.]

“Introduction”

Comment: Should identify more current statistics on worldwide rates of prostate cancer deaths.

Answer: We agree and added the required information and references. [See in Introduction]

Comment: It is unclear how “sociocultural realities that render prostate disease a taboo subject” is ameliorated or even considered in the context of this intervention.

Answer: We agree and removed this aspect from the manuscript.

Comment: It is also unclear how this intervention has the potential to impact limited knowledge of prostate disease that are often not seen until presented complicated or as advanced disease. This is not a community intervention on awareness, but rather a clinical intervention after clinical presentation.

Answer: We agree and make this distinction clear. We also broke down the results to referral groups. We further note in the revision that increasing the public awareness of prostate disease and improving the management of the disease (including patient-physician communication, the subject of this paper) must go hand in hand. [See in Discussion / Relationship to public awareness of prostate disease]

Comment: This authors must reconsider what is important to present as relevant background information for this study.

Answer: We agree and significantly revised the background. [See in Introduction]

Comment: Given the preliminary and exploratory nature of this study, this should be stated clearly upfront. 

Answer: We agree and made clear the preliminary and exploratory nature of this study. We also changed the abstract to reflect this aspect. [See in Abstract and in Introduction]

“Methods”

Comment: Given the discussion of the 3D digital anatomy model section, it is unclear if a standard image was shown to all patients in the study, or if the image was personalized based on the individual patient case. It is unclear why there is a section on a specific patient case of a 62 year old man.

Answer: The MRI “image” was not personalized based on the individual patient case, and we clarify this important detail in the revision. [See in Methods]. We further note that in Senegal, there is a general lack of imaging capacity, especially of MRI. For future work, we consider creating several anatomical models, and based on a quick digital rectal examination, the physician could select the one that feels the most similar one to the prostate of the given patient. [See in Discussion / Limitations / Choice of the 3D anatomical model]

Comment: Under study design, authors indicate “patient first received … standard manual drawings, followed by … 3D digital…” What is the standard manual drawing based on? Is it just what the physician was able to draw themselves or a predrawn out image on a paper? More detail is needed for clarity.

Answer: In their “conventional patient-physician communication”, the participating physicians used didactic anatomical pictures of their own design and drawing, according to their own usual practice. We did not provide a “standardized” image or drawing. We clarified this in the revision. [See in Methods]

Comment: The questions are quite limited in nature and should be addressed in a limitations section.

Answer: We agree, with stressing, as the reviewer noted, that this was a single-arm prospective qualitative study of preliminary and exploratory nature. The questionnaires were purposefully designed to be simple and straightforward, for reasons we discuss in the revised text. [See in Discussion / Limitations / Choice of questionnaires]

 Comment: What, if any, incentives were provided to study participants – whether patients or providers?

Answer: No specific incentives were offered to study participants, neither patients nor providers. During the informed consent process, however, patients were informed about potential benefits, such as receiving more detailed information about their disease and condition and additional “face time” with their consulting physician. We clarified this in the revision. [See in Methods]

“Results”

Comment: A comparison/usual care group could be much more informative and insightful, along with more detailed questions to assess quality of communication, understanding, communication skills, etc., even for low literacy groups, with low response burden.

Answer: We agree that a two-arm study and more nuanced questionnaires could be more informative – and these we intend to pursue in follow-up work outlined in the revised Discussion section. We reemphasize that the purpose of this study was to gauge the perception and generic practicality of 3D digital anatomical models, while keeping the questionnaire unburdening for all participants. [See in Discussion / Limitations / Study design and in Choice of questionnaires]

Comment: Authors should be more clear that the questions assess perception/perceived point of view. For example, asking about improved level of understanding as improved or declined, is perceived and not objective as asking actual questions that assess knowledge. This should also be acknowledged in a limitations section.

Answer: We agree and clarified the issue in the revision, stressing that this was a single-arm prospective qualitative study of exploratory nature, we chose to gauge the perception of the participants, rather than to assess their knowledge and experience gained through using of 3D digital anatomy models. We reserved this objective for future work [See in Discussion / Study Design]

“Discussion”

Comment: Authors should rephrase discussion section and conclusions drawn to be more accurate. For example, quality was not assessed and the conclusion that quality of communication improved is not accurate. There was no comparison group to show an improvement, nor a pre/post questionnaire, for example.

Answer: We agree and clarified in the revision: this was a single-arm prospective qualitative study of exploratory nature, in which we did not have different study arms to compare. We detail this among the Limitations in the revised Discussion session. [See in Discussion / Limitations / Study design and in Choice of questionnaires]

Comment: Impact on physicians – There is also extreme over simplification of communication skills. Nothing that was presented in the intervention dealt with communication – e.g., what is actually said/communicated to patients. This 3D image is a visual “tool” that has the potential to enhance communication, but that was not measured in this study, but rather participants’ perspective on usability or utility, and even that is quite limited in the 4 questions asked of patients and providers.

Answer: We agree and clarify in the revision, as the reviewer correctly noted earlier, that this was a preliminary and exploratory study, in which we did not intend to dwell in a quantitative analysis of such outcomes. The reviewer identifies an important objective for follow-up work that we outlined in the revised Discussion section. [See in Discussion / Limitations / Study design and in Choice of questionnaires]

Comment: Statistical significance – It is unclear why there is a section on statistical significance when this study was exploratory in nature and only presented descriptive statistics. This section seems inappropriate.

Answer: We agree and revised the statistical considerations accordingly. [See in Results / Statistical Considerations]

Comment: The additional subheadings following the discussion section is confusing, when some appear to be appropriate for the methods section.

Answer: We agree and revised the subheadings accordingly. [See in Discussion]

Comment: Digital software – what is the difference between the two platforms mentioned, besides one being free? Why bring this up as an option?

Answer: We intended to say that, for future work, there is now a newer and simpler free open-source digital anatomy platform that provides the same visual functionality as the software we used in this preliminary work. We clarified this in the revised Discussion section. [See in Discussion / Choice of digital anatomy viewing software]

Comment: The section on longevity of findings also seem inappropriate and inaccurate given the nature of this study.

Answer: We agree and arranged it into a subsection on Robustness of findings over time. [See in Discussion / Robustness of findings over time]

Reviewer 3

The changes addressing these comments are marked by blue color in the revised manuscript.

Comment: The manuscript represent the result of a survey on the use of a 3D digital anatomy model in urology consultation to reflect its effect on improving the communication of physician and patients. An interesting model used in this study which based on the results of the survey demonstrated an improved communication between physicians and patients. However, I do not believe the survey results are presented and discussed properly. in more detail (e.g.consider results in referral groups and).

Answer: We agree, and we carried out substantial revisions to address the issues raised by the reviewer, with emphasizing that this was a single-arm prospective qualitative study of preliminary and exploratory nature. We increased the depth of the presentation of Results and Discussion. [See in Results and in Discussion].

Comment. It is not clear if patients information such as age is collected in the survey. If it has been collected, the results should be reported and investigated in terms of any association with age groups (e.g. elderly patients might gain less benefit from the technology compared to younger patients). Otherwise, it should be mentioned that age of patients are not recorded in survey with a discussion on the reason.

Answer: Patient demographics was not collected in this study. We include this clarification in the revised manuscript. [See in Methods / Study Design and in Discussion / Impact on patients]

Comment: There is no statistical method applied to investigate the result of the survey. Authors need to apply statistical methods to add the uncertainty associated with the reported percentage (e.g. in terms of confidence intervals). 

Answer: We agree, and we extended the statistical analysis in the Results section with confidence intervals in both Tables 1 and 2. [See in Results / Answers to the questionnaires]

Comment: Chi‐square or other relevant tests should be applied to investigate the association of each questionary with available patient groups (referral group, age groups, etc). [The] authors need to report results of the questionaries in the survey by the referral groups and investigate any association between referral groups and each questionary (e.g. different treatment usually are recommended for patients with different group so it would be interesting to consider how patients’ decision has changed regarding their referral group).

Answer: We agree, and in the revised paper we break down the results by various referral groups and analysed the results. Chi-square test of independence was performed to analyze the relation between referral groups and different responses in the questionnaire. Bonferroni correction was applied to compensate for repeated tests. [See in Table 3, in Results, and in Discussion / Impact on Patients].

Comment: Impact on patients in discussion, it is mentioned that “A tool that facilitates the communication and understanding of prostate diseases could be beneficial in improving the quality of care and reducing healthcare costs”. However, it is not well discussed how such a tool could reduce healthcare cost, and I suggest authors to add a discussion justifying this (lines 168‐170)

Answer: We agree and added the required discussion. [See in Introduction]

Comment: They mentioned that the use of such a 3D digital anatomy model is helpful for treatment decision of high‐risk patients. It is not clear how the presented results are supporting such a statement! A clear discussion needs to be added including how the patients decisions are improved by use of such a 3D digital anatomy model.

Answer: We agree and in the revised paper, in Table 3, we break down the results by referral groups including the “high-risk” patient groups. Patients favored the digital model regardless of the referral group, the differences were much more significant in decision-making for “high-risk” patient groups who needed immediate management or treatment of the disease. [See in Table 3 and in Discussion / Impact on Patients].

Comment: Some of subsections in the discussion should be moved and presented in the Method section (i.e. Statistical significance, Choice of questionnaires and Choice of digital anatomy viewing software).

Answer: We agree, and we carried out substantial editorial revisions. We moved “Statistical considerations” to the Results section. At the same time, we kept in the Discussion under Limitations the subheadings on “Choice of questionnaires” and “Choice of digital anatomy viewing software”; we deem these necessary to properly discuss limitations of the study. [See in Results and Discussion/Limitations]

Comment: It is not clear how such a software would be available to be used in the clinical practices if required.

Answer: We included a brief discussion of this issue. [See in Discussion / Limitations / Choice of Choice of digital anatomy viewing software]

---

## [Decision Letter · Decision Letter 1]

27 Oct 2022

The use of 3D digital anatomy model improves the quality of communication with patients presenting with prostate disease : first experience in Senegal

PONE-D-21-12063R1

Dear Dr. Diao,

We’re pleased to inform you that your manuscript has been judged scientifically suitable for publication and will be formally accepted for publication once it meets all outstanding technical requirements.

Kind regards,

Yann Benetreau

Staff Editor

PLOS ONE

Additional Editor Comments (optional):

Please note reviewer 3's last request, which you may address in your final version.

Reviewers' comments:

Reviewer's Responses to Questions

**Comments to the Author**

1. If the authors have adequately addressed your comments raised in a previous round of review and you feel that this manuscript is now acceptable for publication, you may indicate that here to bypass the “Comments to the Author” section, enter your conflict of interest statement in the “Confidential to Editor” section, and submit your "Accept" recommendation.

Reviewer #3: All comments have been addressed

2. Is the manuscript technically sound, and do the data support the conclusions?

Reviewer #3: Yes

3. Has the statistical analysis been performed appropriately and rigorously? 

Reviewer #3: Yes

4. Have the authors made all data underlying the findings in their manuscript fully available?

Reviewer #3: (No Response)

5. Is the manuscript presented in an intelligible fashion and written in standard English?

Reviewer #3: (No Response)

6. Review Comments to the Author

Reviewer #3: In the revised version, the authors substantially improved the presentation of results and discussion. I find the revised version addresses my points, and I only have one minor comment:

While I find Table 3 informative and discussed its results well, I suggest that all chi-square test p-values be reported (regardless of whether they are statistically significant).

7. PLOS authors have the option to publish the peer review history of their article (what does this mean?). If published, this will include your full peer review and any attached files.

Reviewer #3: No

---

## [Editor Report · Acceptance letter]

8 Nov 2022

PONE-D-21-12063R1 

The use of 3D digital anatomy model improves the communication with patients presenting with prostate disease: the first experience in Senegal 

Dear Dr. Diao:

I'm pleased to inform you that your manuscript has been deemed suitable for publication in PLOS ONE. Congratulations! Your manuscript is now with our production department. 

Kind regards, 

on behalf of

Dr. Yann Benetreau 

Staff Editor

PLOS ONE